# Prevalence of ADHD in a Sample of Heroin Addicts Receiving Agonist Treatment—Study Conducted in a Public Addiction Service

**DOI:** 10.3390/ijerph20032602

**Published:** 2023-01-31

**Authors:** Pasqualina Rocco

**Affiliations:** Addiction Treatment Center, Local Health Service N. 2, Veneto, Via dei Carpani, 16/Z, 31033 Treviso, Italy; pasqualina.rocco@aulss2.veneto.it

**Keywords:** addiction, ADHD, agonist treatment, heroin addicts, opioid addiction

## Abstract

Attention-deficit/hyperactivity disorder (ADHD) is a childhood neurodevelopmental disorder that can persist into adulthood. The co-occurrence of ADHD and substance use disorders is very frequent and has received considerable attention in recent clinical/scientific investigations. However, few studies have investigated the prevalence of ADHD in heroin addicts. This study aimed to investigate the prevalence of attention-deficit/hyperactivity disorder (ADHD) in a sample of heroin addicts treated with opioid agonists and to report this clinical experience in a public service for addiction. Outpatients over 18 years old and being treated with opioid agonists for heroin addiction were enrolled. Each patient took part in a psychiatric examination and completed an ASRS (Adult ADHD Self-Report Scale) self-assessment. Subjects with positive results were called in for another psychiatric visit, and the Brown ADD scale was used as a second-level test for ADHD; furthermore, the Mini International Neuropsychiatric Interview (MINI) and Hypomania/Mania Checklist (HCL-32) were used for differential diagnoses and to assess comorbidities. In total, 111 patients were enrolled. All were followed up by the psychiatrist, who is also the author of this report and the person who formulated the diagnoses. The prevalence of ADHD in this sample was 18%. Among the 20 patients diagnosed with ADHD, 5 (25%) were female and 15 (75%) were male. The most frequent psychiatric comorbidity was major depression, found in 11 patients (55%), of which 4 presented with hypomania (bipolar disorder). In this sample, making diagnoses was very difficult. Frequently, multiple comorbidities further complicated these cases. In conclusion, the results of this study are consistent with the literature: There seems to be a significant prevalence of ADHD even among heroin addicts, and often, the diagnosis is difficult to make. We also do not know the exact effect of opioid agonist therapy on ADHD symptoms. Hypotheses have been put forward, but studies are needed.

## 1. Introduction

Attention-deficit/hyperactivity disorder (ADHD) is a childhood neurodevelopmental disorder characterised by inattentiveness and impulsivity, which can be associated with hyperactivity; it also persists into adulthood in two-thirds of cases [1].

The population prevalence of adult ADHD is reported to be 3–5% [2] (pp. 402–409), [3].

The prevalence of psychiatric disorders (other than ADHD) is higher in adults with ADHD than in the general population [3,4]. According to several studies, psychiatric comorbidity in adult ADHD ranges from 60 to 80% [5] (pp. 210–2016). The most commonly reported comorbid disorder is drug abuse [6] (pp. 291–296), [7] (pp. 84–93), [8] (pp. 1066–1075).

The co-occurrence of ADHD and substance use disorders has received considerable attention in recent clinical and scientific investigations because it is very frequent and has serious clinical implications [6] (pp. 291–296).

In drug addicts, the prevalence of ADHD is estimated to range from 14 to 44% [9] (pp. 601–608), [10] (pp. 158–166).

The ADHD–drug addiction comorbidity is associated with multiple problems, including:Increased psychiatric comorbidity [11] (pp. 262–272);Increased risk of relapse (even for the greatest dyscontrol) [12] (pp. 1875–1882), [13] (pp. 1057–1063);Longer duration and greater severity of drug addiction [14] (pp. 683–693);Greater polysubstance use [15] (pp. 43–51); andLower effectiveness of treatments [16] (pp. 15–23).

Studying this type of patient is difficult, and this difficulty emerges directly or indirectly in many of these studies, starting with the diagnosis.

Regarding the type of substances used by ADHD patients, in the literature, a robust association between ADHD and nicotine and alcohol use disorders is reported [17] (pp. 9–21).

If we specifically focus on opioid use disorder, few studies have reported the prevalence of ADHD [18] (pp. 164–171), perhaps because ADHD patients have historically been known to use stimulant drugs. However, these few studies seem to indicate that, even among heroin addicts, there is a good percentage of people who suffer from ADHD. 

King et al. found a 16.7% prevalence of adult ADHD in a group of patients entering methadone maintenance treatment [19] (pp. 487–495).

In some studies conducted with methadone patients and others with opioid-use disorders, the rate of prevalence of adult ADHD diagnosis was about 24% [20] (pp. 10–20), [10] (pp. 158–166), [11] (pp. 262–272). The role of agonist therapy is not clear in these studies.

In a study by Cuneyt E. et al. in 2018, 234 heroin addicts were analysed. The ASRS, SCL-90, and the short form of the Barratt Impulsiveness Scale (BIS-11-SF) assessments were used. The authors found that about 23% of the participants were likely to have ADHD. However, the aim of the study was to correlate ADHD with psychopathology and impulsivity. The authors reported that, in ADHD patients, substance use disorder may facilitate impulsive acts. The authors did not specify whether self-harm also increases in these patients, especially if they also have mood disorders [18] (pp. 164–171).

In 2017, an Italian multicentric study was published in which 1057 heroin addicts in opioid treatment completed the ASRS for ADHD and the SCL-90. About 19% of the participants screened positive for concurrent adult ADHD symptoms. Unfortunately, this study only evaluated ADHD symptoms (which are frequent in drug addicts) and did not verify the ADHD diagnosis [5] (pp. 2010–2016).

This Italian study, conducted with patients in the public addiction service, has the merit of focusing attention on this type of patient.

The author of this study also works in a public addiction service in which the majority of patients are addicted to heroin. Clinical observation often leads to the suspicion of ADHD, but there are no specific and shared protocols to identify such subjects. It is important to identify these patients so as to improve the management of their addictions. For this reason, a protocol very similar to that used in normal clinical practice was devised, at a low cost and with immediate results.

For practical reasons, the need arose in the clinical context, and not in a research university, to evaluate the prevalence of ADHD in heroin addicts being treated with opioid agonists.

The purpose of this report is to detail this experience and the results so as to give food for thought and research.

## 2. Methods

This study was conducted at a public addiction treatment centre (Local Health Service n.2) in Castelfranco Veneto, Italy. As mentioned in the introduction, it is a public centre of the National Health Service. These centres deal with addiction and are multiprofessional (employing doctors, psychologists, nurses, educators, social workers, etc.). This service was implemented in Italy in the 1970s–1980s when heroin became a major problem; they are today rather well-regarded in this sense.

Castelfranco’s service only operates on an outpatient basis, and those receiving treatment are predominantly heroin addicts.

The subjects in this sample had the following characteristics:

Adult outpatients diagnosed with opioid use disorder according to the criteria listed in the *DSM-5* [21].

All patients had been on maintenance therapy (methadone, levomethadone, or buprenorphine) for at least 6 months. Prior years of treatment ranged from 6 months to 30 years. Most of these patients had been engaged in therapy for at least 10 years (44.1%).

No participant was being treated with ADHD therapy (methylphenidate or atomoxetine), either before or during agonist treatment.

None of the participants had been diagnosed with ADHD in childhood. (It is known that the disorder in Italy, as in some other countries, appears to be underestimated, even in childhood. This appears to be one of the main reasons why very few patients in public addiction services have a prior diagnosis of ADHD).

All of the patients in this centre, without exception, entered into treatment voluntarily and signed an informed consent form, as is a part of our normal practice.

Psychiatric visits and the prescription of therapy are part of the routine practice of the service. The tests used in this study are also routinely used in our clinical practice. They were only used more systematically for the purposes of this study. As already mentioned in the introduction, the study was designed to be as similar as possible to our normal clinical practice, with a very low cost, easy application, and immediately accessible data.

All of the enrolled patients received a psychiatric visit, and the Adult ADHD Self-Report Scale (ASRS v1.1) was administered [22]. The visiting psychiatrist explained the test to the patient and assisted them in its completion.

The first test used was a screening test: the Adult ADHD Self-Report Scale (ASRS v1.1):

The Adult ADHD Self-Report Scale (ASRS v1.1) is an 18-item self-report questionnaire designed to assess attention-deficit/hyperactivity disorder (ADHD) symptoms in adults (18+ years of age). This scale is based on the World Health Organization Composite International Diagnostic Interview (2001), and the questions are consistent with *DSM* criteria, but they have been reworded to better reflect symptom manifestation in adults. This scale is useful for the screening and diagnosis of ADHD among adults above 18 years of age and should be used in conjunction with a clinical interview to provide additional clinical information.


*“Part A: contains 6 items and it has been found that these questions are the most predictive of ADHD and are best for use as a screening instrument. Part-B: contains 12 additional questions based on DSM criteria that provide additional cues and can serve as further probes into the patient’s symptoms. For a client’s symptoms to be considered consistent with an ADHD diagnosis, they require 4 or more responses at specific severity levels in Part A of the ASRS”*
[22] (pp. 245–256)

The ASRS is considered a good screening scale (and it is also used in drug addicts); it has been validated and is widely used in clinics and in research studies [23] (pp. 299–305), [22] (pp. 245–256).

Patients who were judged to be positive for ADHD, according to the ASRS and/or clinical observation, continued the study.

Another psychiatric visit was made for better diagnostic investigation. Whenever possible, the psychiatrist also collected anamnestic information from family members and school documentation with the teachers’ evaluations. He paid particular attention to the patient’s functioning prior to substance use.

Another assessment was then administered: the Brown ADD scale (Brown Attention-Deficit Disorder Scale) for adults [24].

The Brown ADD scale for adults evaluates symptoms attributable to attention-deficit disorder (ADD). It consists of several sections, including an anamnestic part and a psychiatric comorbidity part, that are administered via semi-structured interviews. It also includes a self-administered section of 40 items and the comparison of the patient’s answers with those of a family member or acquaintance. The self-administered items can be grouped into six clusters:(1)Organisation, establish priorities;(2)Focus, support, and shift attention;(3)Regulation of activation, support the effort;(4)Manage frustration and modulate emotions;(5)Working memory and re-enactment; and(6)Monitor and self-regulate the action.

An explanation of these parameters is contained in the test manual, and these parameters were applied in the present study [24].

Brown-scale-positive patients were then given other tests for the differential diagnosis: the MINI and the HCL-32.

The Mini International Neuropsychiatric Interview (MINI v5.0.1) is a diagnostic rating scale. It includes a semi-structured interview compatible with *DSM-4* criteria, and it is used both in clinics and in research studies [25]. The interview is divided into modules, with each corresponding to a diagnostic category (*DSM-4*), and at the end, the clinician can indicate whether a category has been satisfied. The diagnostic categories considered in the test are as follow: major depressive episode with or without melancholia, dysthymia, suicide risk, (hypo)maniac episode, panic, agoraphobia, social phobia, generalized anxiety disorder, obsessive–compulsive disorder, posttraumatic stress disorder, and substance addiction, and there is an optional antisocial personality module.

The Hypomania/Mania checklist (HCL-32) is a self-rating questionnaire designed to assess a lifetime history of hypomanic symptoms [26] (pp. 210–233). It is a comprehensive and simple questionnaire used to screen for bipolar disorder. It consists of 32 questions, and its purpose is to assess the characteristics of the respondent’s “high” periods.

All of the tests used have been validated.

We know that the diagnosis of ADHD is clinical and can be supported by validated tests. Therefore, our working method followed from this knowledge.

The scheme is summarised as follows: All enrolled patients received a psychiatric visit and the ASRS v1.1. ASRS-positive patients continued in the study. ASRS-negative patients continued in the study if the psychiatrist suspected that their scores were false-negatives. Otherwise, those scoring negatively did not continue in the study.

The ASRS-positive participants, as well as those who scored as ASRS-negative but who were suspected to have scored as false-negative, were asked to participate in another psychiatric visit. The Brown scale (a second-level diagnostic test) was administered; the MINI and HCL-32 were also administered so as to aid in the differential diagnosis. This ended the diagnostic process. A control group was not foreseen. This study aimed to observe the prevalence of ADHD in a common clinical practice procedure. 

All psychiatric visits and testing were performed by the same psychiatrist.

## 3. Results

In total, 111 patients participated in the study; 89 were male (80.1%), and 22 were female (19.8%). The average age of the participants was 38.9 years.

The following socio-demographic parameters were taken into consideration: sex (m/f);

age;marital status (single/married/divorced or separated);educational attainment (primary/lower secondary/secondary–university);employment (stable, occasional, unemployed, unable to work);cohabiting (parent, partner, friends, community, alone); andclarification: lives with children (yes/no),Table 1 describes the characteristics of the sample.

It can be seen that this sample was mostly made up of single people with a low–medium educational level. Most had stable jobs and lived with their parents.

Types of secondary substances used were also taken into consideration, as reported by patients and confirmed by urinary toxicology tests.

All patients smoked cigarettes (tobacco) (100%).

The second-most frequently used substance was cocaine, alone (24.3%) or with the same frequency as another substance (cannabis 15.3%, alcohol 2.7%). This affected almost half of the patients (42.3%).

Cannabinoid use was also frequent throughout the sample alone (19.8%) or with other substances (cocaine 15.3%; alcohol 2.7%), totalling 36%.

As one can see, alcohol was often associated with the use of other substances; alone, it affected 8.1% of patients.

The use of amphetamines and MDMA were infrequent in this sample: 1.8%.

A total of 4.5% of patients reported engaging in polysubstance abuse, i.e., the abuse of more than two secondary substances.

A total of 22.5% of the patients were in treatment with agonist therapy without a secondary substance (or with occasional use of other substances).

Of the 111 patients, 83 (74.4%) scored negative on the ASRS. However, 15 of these were suspected of having ADHD (probable false-negatives) based on clinical observation, and they continued in the study.

A total of 68 of 83 patients were excluded because they were assessed as negative for ADHD.

A total of 28 patients (25.2%) out of the 111 respondents tested positive on the ASRS, and they continued on to the second phase of the diagnostic study. However, 6 ASRS-positive patients failed to enter the second phase due to dropout. Thus, 22 patients (19.8%) remained. These 22 positive-ASRS patients, combined with the 15 suspected false-negative patients, made up a total of 37 patients who underwent the second part of the study.

These 37 patients participated in another psychiatric visit, and they were administered the Brown ADD scale. Six patients tested negative on this assessment. Another 11 tested negative but borderline for ADHD; that is, some requirements were missing, but they had many ADHD characteristics.

Finally, 20 patients tested ADHD positive (18% of the initial sample).

Among the 20 patients diagnosed with ADHD, 5 (25%) were female, and 15 (75%) were male.

The scheme of study and the results are shown in Figure 1.

(Figure 1: Scheme of study and results. The figure shows the scheme of this work: it began with the administration of the ASRS to 111 patients. Then, two groups, the suspected false-negative ASRS group and the positive ASRS group, continued on to the administration of the Brown scale. The results are shown; the numbers and percentages of the patients are also reported).

These 20 patients were ten administered the MINI and HCL-32, along with participation in a confirmatory psychiatric visit.

The most frequent comorbidity was lifetime major depression (55%); 20% of these cases came with a diagnosis of bipolar disorder; 35% showed lifetime major depression without a diagnosis of bipolar disorder. Anxiety–panic disorders affected 35% of the patients. The diagnosis of antisocial personality disorder was associated with 25% of the cases.

The MINI also assesses the risk of suicide. The risk was present in 20% of these patients.

A total of 40% of patients had at least two other psychiatric diagnoses in addition to the diagnoses of substance use disorder and ADHD.

## 4. Discussion

This study was conceived in a clinical context, not in a research centre or a university. This centre treats many heroin addicts. In our clinical practice, ADHD symptoms are very frequent; some research has confirmed this observation. 

For us, it is very important to understand whether these symptoms correspond to a diagnosis of ADHD for practical reasons: we need to plan more targeted interventions and therapies. For this reason, a protocol was designed that is practical and relatively easy to apply so as to understand how many patients actually present with a diagnosis of ADHD.

The diagnostic study of ADHD in drug-addicted patients is very difficult for several reasons. One of these is that the presence of substance use can induce secondary ADHD symptoms and confuse the clinical presentation. Substance use alters cognitive modalities, causes attention deficits, and affects cognitive flexibility, working memory, and impulsivity management, similarly to ADHD [27]. The difference is that ADHD symptoms will have been present since childhood, but this is not always easy to ascertain.

We must take this into account, but it should not stop us from looking for an answer. We have a duty to do this for these patients.

In this study, an attempt was made to enrol a slightly more stable category of patients. Among drug users, those who use heroin as their primary drug and are receiving agonist treatment generally appear more stabilised. From this group, patients who had been in treatment for at least 6 months were chosen. Moreover, almost all of these patients were in the stabilisation phase at the time of observation; only a few were in the disengagement phase (on discharge from the service).

The presence of agonist therapy and the treatment phase were chosen to try to eliminate as much interference from the use of substances as possible; this interference could not, of course, be ruled out, but at least it was reduced.

The psychiatric examination of the patients was conducted by the same physician for the duration of the study; this was the same doctor who treats these patients in the facility and knows them. This ruled out the possibility of a double-blind study, but it was preferred as an aid to the differential diagnosis.

The patient sample was not randomised, and it has particular characteristics. The patients belong to a public, outpatient facility. In Italy, there is a very widespread public treatment network, and most of the patients who use heroin enter the public service. Only the public service can provide agonist therapy to these patients. The choice of patients treated with agonists allowed access to a large sample of heroin addicts. However, it must be taken into account that participation in this study was voluntary; it is likely that has resulted in the selection of less-severely affected patients.

Another limitation of this study is that not all enrolled patients were followed up to the end. This prevented the comparison of the positive and negative ADHD groups.

As far as the tests are concerned, they were chosen from those that are most commonly known and used in Italian psychiatric and addiction services. The ASRS is a very common and easy-to-use screening scale; the Brown scale is among the most-used for diagnosis in second-level testing.

For the differential diagnosis, we chose the MINI, which includes all of the main psychiatric diagnoses and, above all, it is easy to use.

The HCL-32 scale, which is also very practical, is used for the diagnosis of bipolar disorder, which is often associated with, or can be confused with, ADHD.

At the end of the diagnostic process, an ADHD frequency of 18% was found. It is a frequency compatible with the data as published in the literature; this encourages us to think that this method could be effective in clinical practice.

Our results seem to confirm that, even among heroin-using patients, there is a significant percentage of people with ADHD, and this is very important for the prognosis, course, therapy, and organisation of services.

Interestingly, there were six dropouts among the ASRS-positive patients during the study. This number corresponds to 5.4% of the total sample, but also to 30% of the ASRS-positive sample. We do not have a ASRS-negative group comparison; nonetheless, we expect ASRS-positive individuals to be more impulsive and more severely affected, so they would be more likely to exit the program. However, we think that this could be a useful field for future investigation.

We report that it was very difficult to diagnose ADHD in these patients for several reasons.

We know that ADHD is more difficult to diagnose in adults, notably because the presentation of symptoms changes in adulthood. The presence of substance use disorder seems to be a further complication.

We had diagnostic difficulties mainly due to:The effect of the substances, as stated above (symptoms secondary to substance use).The anamnestic reconstruction. (For example, we requested school reports from elementary schools. Some, but not all, patients managed to produce their school reports. In Italy, the teachers seem very good at reporting difficulties with concentration and attention, and this proved to be a helpful resource).The collaboration of family members. (Not all patients had cooperative or reliable relatives).The patients’ own recognition of symptoms. ADHD patients usually have little insight. This was a very obvious problem. All of the patients in the study were assisted in completing the self-administered tests. They were assisted by the same psychiatrist who had them in his charge, and he knew them well. The psychiatrist saw that they did not recognize their symptoms on the test, and it was often necessary to cause them to reflect and give examples. On the other hand, the presence of a psychiatrist who knew the patients could have influenced the results.High comorbidity. Other disorders may alter the presentation of symptoms.

In evaluating patients with borderline symptoms of ADHD (those who did not meet all criteria for a *DSM* diagnosis and/or had borderline test results), we often wondered if agonist therapy might be playing a role. We know that it stabilises heroin addiction, exerts an anticraving effect on other substances, and has other effects on psychopathological symptoms (e.g., anxiety or psychotic symptoms); however, we do not know if it also stabilises ADHD symptoms by making them weaker.

We have discussed this with ADHD patients who are on agonist treatment. They report the improvement of symptoms with therapy. This is a very interesting point on which, unfortunately, the literature is still very scarce.

With regard to psychiatric comorbidity, the data are compatible with the literature (see introduction) [11]. We noted that there is a lack of cases of association with schizophrenia. We think that it is mainly due to two reasons: (1) too small of a sample size. (There are not many schizophrenics enrolled in these services, so a very large sample would be needed.); (2) patients who, due to their characteristics, refused to enter the study.

Another unsurprising figure is the high percentage of patients who were assessed as being at a risk of suicide on the MINI (20%). We know that ADHD patients display less self-control, and this facilitates self-harm. Additionally, the risk of suicide in these patients seems to have been considered rarely in the literature. In our opinion, this topic deserves further study.

## 5. Conclusions

This study was conducted in a public addiction service where there was a need to diagnose ADHD patients for better clinical practice. The literature is limited, so a protocol with resources and tools normally available in the service was applied. The purpose of this report was to demonstrate this clinical practice experience and to show the results of the prevalence of ADHD in these patients. The intent was also to raise awareness of the problem and provide ideas for future research.

In this sample, the prevalence of ADHD was found to be 18%; this result is consistent with that found in the literature. The psychiatric comorbidity data are also compatible with those found the literature. Affective disorders appeared to be the most frequently associated psychiatric disorders in this sample. Multiple comorbidities very frequently appeared, and there was a high percentage (20%) of subjects who were at risk of suicide.

We report that it was very difficult to diagnose ADHD in these patients. Strategies were sought to help with the anamnesis and to support the poor insight of the patients. An attempt was made to reduce the interference of substances. However, the problem of diagnosis remains difficult to solve.

We have also brought attention to the role of agonist therapy, asking the question of whether it can improve symptoms associated with this disorder. It would be important to understand this therapy better, which does not yet seem to have been taken into consideration in the literature.

Another focus was placed on the suicide risk of these patients, a worrying fact that deserves more attention. The risk of suicide in drug addicts has rarely been studied. In theory, ADHD-affected drug addicts, often having lower self-control, should be at even greater risk. However, this hypothesis needs to be proved.

The difficulties of studying these patients, at the level of clinical practice and even more so in systematic research, must not stop us. On the contrary, these difficulties must stimulate us even more to seek answers.

## Figures and Tables

**Figure 1 ijerph-20-02602-f001:**
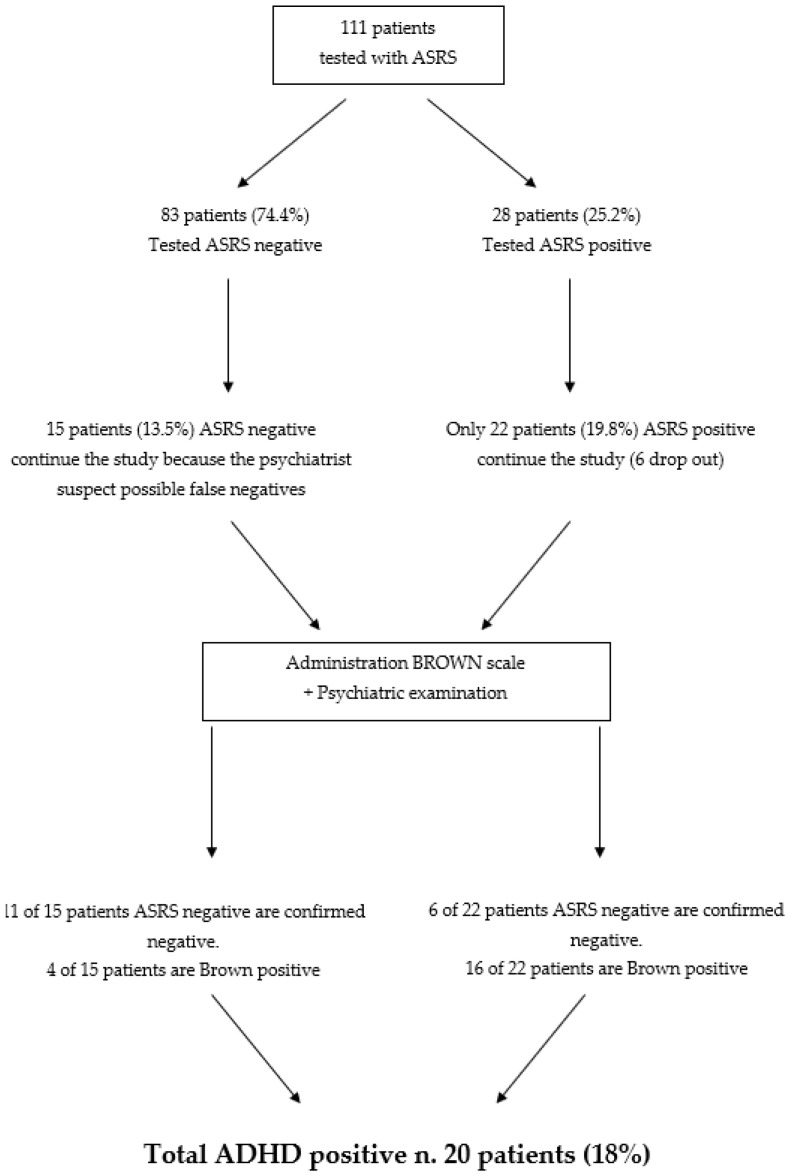
Study Scheme and Results.

**Table 1 ijerph-20-02602-t001:** Demographic and socio-economic characteristics of the sample, with percentages.

Demographic/Socio-Economic Characteristics	%
Marital status	Single	74.7
	Married	17.1
	Divorced	8.1
Educational attainment	Primary	2.7
	Lower secondary	65.7
	Secondary/Univ.	31.5
Employment	Stable	61.2
	Occasional	9
	Unemployed	24.3
	Unable to work	5.4
Cohabiting	Parents	56.7
	Partner	27.9
	Friend	2.7
	Community	1.8
	Alone	10.8
Lives with children	16.2%

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
