# Peer review of "Prevalence of ADHD in a Sample of Heroin Addicts Receiving Agonist Treatment—Study Conducted in a Public Addiction Service"

_ijerph, 2023, doi:10.3390/ijerph20032602_

Round 1

Reviewer 1 Report

Interesting topic; research well built and well structured narrated.

No comments except that it should be DSM 5 and not DSM V

Author Response

Point 1: No comments except that it should be DSM 5 and not DSM V

Response 1: DSM V has been changed to DSM 5. 

I also asked the journal for help in reviewing the English language.

Reviewer 2 Report

This manuscript focusses Attention deficit hyperactivity disorder (ADHD and drug addiction. Even though this topic is very interesting for the scientific community, I cannot recommend its publication. As I appreciate the work of the authors, I would like to list my reasons for my decision below.

I read this manuscript several times and I was slightly confused as it appears that it has not been completed. It appears to be very fragmented and lacks methodological descriptions etc... I really appreciate the work what the researchers put in in this manuscript. However, I suggest that in its current form it should not be published; and I also have to say that it goes beyond the scope of a reviewing process to revise the manuscript for now. Since I believe that the work is crucial and covers important issues on ADHD and drug addiction, I would like to encourage the authors to improve the entire manuscript (structure, writing, content etc.) including analyses. As I want to be kind I will spend some time on giving feedback, even though I do not recommend its publication.

General remarks:

Why are some passages highlighted in colour? Unless this is not a format which is provided by the journal, I find it rather confusing. It appears as if some parts were copied from another text etc….

Language:

 you often have 1 sentence paragraphs, and the text is incoherent – this needs to be changed. A paragraph should consist of more sentences which deal with a particular aspect……

Abstract:

Some issues

Line 13 
“utpatients..” Is there something missing?

Line 18 – in the text is says “111” here it says “11” – please check these things

Line 19/20 you need to mention who diagnosed ADHD and also check the percentages you describe in the abstract

Some issues:

Introduction:

I find the text rather unfragmented and too short for an introduction. Please include more information. Also, you need to provide the reader for what is going on in the discussion in the introduction before. There needs to be a connection to the two parts. This means that information which appears in the discussion should already be provided in the introduction to help the reader to understand your rational idea in the end. The same may be for methods ( e.g. what other researchers did who work on similar topics….)

Many formatting problems – size of the letters change and sometimes it is blue, sometimes grey etc… appears as if it was copied from another passage

Methods

In general – there are many more details needed. I give you an example below:

 It is not enough to present provide key terms of the test. E.g. working memory and re-enactment – there needs to be description about what these measures show/ provide etc… researchers who read this need to be able to compare methods to other work. ….

Figures: Quality  - the figures should be of better quality

Results:

I think that there are better analyses should be provided and not only the percentages and number of each patient. Statistical analysis  is probably required.

Discussion:

You need to link the discussion to the introduction.  

Line 236 – figure – why is this figure in the discussion section and not in the results section?

Author Response

Point 1: I read this manuscript several times and I was slightly confused as it appears that it has not been completed. It appears to be very fragmented and lacks methodological descriptions etc... I really appreciate the work what the researchers put in in this manuscript. However, I suggest that in its current form it should not be published; and I also have to say that it goes beyond the scope of a reviewing process to revise the manuscript for now. Since I believe that the work is crucial and covers important issues on ADHD and drug addiction, I would like to encourage the authors to improve the entire manuscript (structure, writing, content etc.) including analyses. As I want to be kind I will spend some time on giving feedback, even though I do not recommend its publication.

Response 1: In fact, the piece has printing and formatting errors, due to the rush to send it to the deadline. For the same reason it seems incomplete.  In these things the work has been improved. Fragmentation and methodological descriptions have also been improved.

Then it is necessary to consider the context where this report was born and its purpose. The report describes a clinical experience and not a university research. It wants to show a type of pratical work  and suggest ideas to researchers. I think that sometimes interesting intuitions can arise from observations.  After your remarks I tried to explain it better in the text. 

The English language has been revised by the journal (certificate). The whole report has been much improved. I believe that with the right premises and the changes made the report deserves publication.

Point 2: General remarks:

Why are some passages highlighted in colour? Unless this is not a format which is provided by the journal, I find it rather confusing. It appears as if some parts were copied from another text etc….

Response 2: some passages are highlighted in color by mistake. A uniform color has been applied. There are no copies, only pieces of speech translated and reported at different times. This report was originally a poster, always written by me.  The poster was presented at the 2nd Word Congress of World Association on Dual Disorders in Italy in 2018. It was deemed interesting and the congress organizers encouraged its publication. So the text was integrated to make a report. We hastlily prepared for the deadline. The only copied text is the explanation of the ASRS scale and has been highlighted with a different graphic character.

Point 3: Language:

you often have 1 sentence paragraphs, and the text is incoherent – this needs to be changed. A paragraph should consist of more sentences which deal with a particular aspect……

Response 3: the paragraphs are a formatting error. Have been formatted.

Point 4: Abstract:

Some issues

Line 13 
“utpatients..” Is there something missing?

Line 18 – in the text is says “111” here it says “11” – please check these things

Line 19/20 you need to mention who diagnosed ADHD and also check the percentages you describe in the abstract

Response 4: printing errors have been corrected. I specified who diagnoses ADHD, the percentage were checked. I also revised the sentence of the purpose.

Point 5: Introduction:

I find the text rather unfragmented and too short for an introduction. Please include more information. Also, you need to provide the reader for what is going on in the discussion in the introduction before. There needs to be a connection to the two parts. This means that information which appears in the discussion should already be provided in the introduction to help the reader to understand your rational idea in the end. The same may be for methods ( e.g. what other researchers did who work on similar topics….)

Many formatting problems – size of the letters change and sometimes it is blue, sometimes grey etc… appears as if it was copied from another passage

Response 5: the introduction has been revised. It certainly remains concise and essential. You have to remember that this is a short report and not a review of the literature. For this reason it only reports the salient aspects of the problem. As requested, the introduction was better connected to the discussion. What other authors have done on similar topics has been explained and the purpose of this report has also been clarified. The formatting has been corrected. There are no copied parts (see response 2).

Point 6: Methods

In general – there are many more details needed. I give you an example below:

 It is not enough to present provide key terms of the test. E.g. working memory and re-enactment – there needs to be description about what these measures show/ provide etc… researchers who read this need to be able to compare methods to other work. ….

Figures: Quality  - the figures should be of better quality

Response 6: many more details have been provided in the "Methods". As for the description of key terms of the test, for example "Working memory", these are all explained in the test manual. You have to stick to those meanings. However, I have clarified this in the report. Opening a discussion on these terms as understood by the test, would be long and complex and it is not the purpose of this report. The figures have been improved following the indication of the journal.

Point 7 Results:

I think that there are better analyses should be provided and not only the percentages and number of each patient. Statistical analysis  is probably required.

Response 7: we must remember that we are in a context of clinical practice and not university research. Moreover, the aim is to show what is done in a public service to arrive at the diagnosis. I added this clarification in the discussion. I have eliminated the figure on the use of secondary substances and the comparison between the two groups (ADHD positive and negative), because it is beyond the scope of prevalence. The rest of the study is just an observation, it has no case-control for statistics. I explained this too.

Point 8: Discussion:

You need to link the discussion to the introduction.  

Line 236 – figure – why is this figure in the discussion section and not in the results section?

Response 8: the discussion was linked to the introduction. The percentage that was on line 236 is also in the results on line 170.  If you are referring to the image (graph), this has been removed as not relevant (see also response 7).

Reviewer 3 Report

This study aimed to investigate the prevalence of attention-deficit/hyperactivity disorder (ADHD) in a sample of heroin addicts treated with opioid agonists.

-Indeed, the relationship between ADHD and addictions is important.

-First I would like to express my concern about the feasibility of the research. I believe that the diagnosis of ADHD in people who are already addicted probably cannot give safe results. Especially considering the sample size. The use of such strong substances can induce various secondary symptoms including those of ADHD. However, I appreciate that the author mentions this limitation in the discussion. Of course, this does not mean that the present research has less value. However, It is essential to support your study’s importance by better describing the objectives, and the reason why this study has added value.  For instance, in the introduction, the objective is described in only two lines (page 2, lines 53-54).

- The way of presentation could be more coherent. I understand that it is about a brief report. However, in several parts of the text, the speech is telegraphic, with small paragraphs. It gives a confusing impression to the reader (i.e., lines 53-54, 61-64, 214-218). Thus, I suggest a better organization of the text.

-Care should be taken in formatting the text, figures, and tables. For figures and tables, it would be better to use the journal’s templates. The second figure does not have a rubric or a description. In addition, in the text, you mention (see figure) without a number.

According to the authors' guidelines: In the text, reference numbers should be placed in square brackets [ ], and placed before the punctuation; for example [1], [1–3] or [1,3]. For embedded citations in the text with pagination, use both parentheses and brackets to indicate the reference number and page numbers; for example [5] (p. 10). or [6] (pp. 101–105).

-For the reference list, ACS style guide is recommended.

Author Response

Point 1:

First I would like to express my concern about the feasibility of the research. I believe that the diagnosis of ADHD in people who are already addicted probably cannot give safe results. Especially considering the sample sizeThe use of such strong substances can induce various secondary symptoms including those of ADHD. However, I appreciate that the author mentions this limitation in the discussion. Of course, this does not mean that the present research has less value. However, It is essential to support your study’s importance by better describing the objectives, and the reason why this study has added value.  For instance, in the introduction, the objective is described in only two lines (page 2, lines 53-54).

Response 1: It's really, really hard to study ADHD in drug addicts. However we have a duty to try, especially for these patients. It is true that substances can induce ADHD symptoms. However, heroin addicts in continuos treatment are generally quite stabilized. Also the accurate anamnesis with documentation and testimonies of the family is useful. These aspects have been brought into discussion. There are not many studies on the subject and there are studios with samples of a few dozen. This report does not, in my opinion, have a small sample. It must be taken into account that the report it is a clinical experience, an observation in a public service, not a university research. Furthermore, its purpose is not so much to provide certaines, but to show clinical experience and a raise useful questions for research. I have tried to explain this better in the introduction, in the objectives and in the discussion.

Point 2:

The way of presentation could be more coherent. I understand that it is about a brief report. However, in several parts of the text, the speech is telegraphic, with small paragraphs. It gives a confusing impression to the reader (i.e., lines 53-54, 61-64, 214-218). Thus, I suggest a better organization of the text.

Response 2: Sometimes the speech is telegraphic because this report was a poster presented to the 2nd World Congress of World Association on Dual Disorders in 2018. Then the poster was processed as a report. The work was sent in a hurry for the deadline. However now the whole report has been improved.

However all the text has been improved and also the suggested lines  (53-54, 61-64, 214-218).

Point 3:

Care should be taken in formatting the text, figures, and tablesFor figures and tables, it would be better to use the journal’s templates. The second figure does not have a rubric or a description. In addition, in the text, you mention (see figure) without a number.

Response 3: the text, table and figures have been formatted. The table and the scheme have been changed, as indicated by the journal. The numbers, titles and explanation have been inserted. The study scheme has been improved.

Point 4:   

According to the authors' guidelines: In the text, reference numbers should be placed in square brackets [ ], and placed before the punctuation; for example [1], [1–3] or [1,3]. For embedded citations in the text with pagination, use both parentheses and brackets to indicate the reference number and page numbers; for example [5] (p. 10). or [6] (pp. 101–105).

-For the reference list, ACS style guide is recommended.

Response 4: citations and references have been modified as required.      
